# Olive Oil Intake Associated with Increased Attention Scores in Women Living with HIV: Findings from the Chicago Women’s Interagency HIV Study

**DOI:** 10.3390/nu11081759

**Published:** 2019-07-31

**Authors:** Lakshmi Warrior, Kathleen M. Weber, Elizabeth Daubert, Martha Clare Morris, Puja Agarwal, Igor J. Koralnik, Audrey L. French

**Affiliations:** 1Division of Neurology, Department of Medicine, Cook County Health, Chicago, IL 60612, USA; 2Department of Neurological Sciences, Rush Medical College, Chicago, IL 60612, USA; 3Core Center, Cook County Health/Hektoen Institute of Medicine, Chicago, IL 60608, USA; 4Rush Institute for Healthy Aging, Department of Internal Medicine, Rush Medical College, Chicago, IL 60612, USA; 5Department of Medicine, Stroger Hospital and Rush University, Chicago, IL 60612, USA

**Keywords:** cognition, HIV, women, olive oil, attention

## Abstract

Women aging with human immunodeficiency virus (HIV) are particularly vulnerable to cognitive decline. Recent studies have highlighted the potential protective effects of olive oil on cognition in persons living without HIV. We sought to evaluate the association between olive oil consumption and domain-specific cognitive performance (dCog) *t*-scores (adjusted for age, race, education, reading level, practice effects) in women living with HIV (WLWH) and sociodemographically similar women living without HIV. A total of 166 women (113 WLWH and 53 women living without HIV) participating in the Cook County Women’s Interagency HIV Study (WIHS) completed cognitive testing and a Block 2014 Food Frequency Questionnaire within 18 months. Use of olive oil was associated with a 4.2 point higher attention/concentration (*p* = 0.02), 4.0 point higher for verbal learning (*p* = 0.02), and 1.91 point higher for verbal memory (*p* = 0.05). Associations between using olive oil and attention/concentration cognitive domain were seen in WLWH but not in women living without HIV. Associations between olive oil and verbal learning and memory were only seen in women without HIV. Our data suggest that using olive oil as a primary cooking oil may contribute to differential effects in attention/concentration, verbal learning, and verbal memory between women living with and without HIV.

## 1. Introduction

Diet has emerged as an important modifiable risk factor for cognitive decline [1,2,3]. The effects of olive oil have been highlighted in recent studies, and supplementation of olive oil is associated with improved performance in specific cognitive domains [1]. The mechanism of neuroprotection for olive oil is thought to be through a combination of anti-inflammatory and antioxidant effects, as well as prevention of cerebrovascular events [4].

With a growing population of aging individuals with human immunodeficiency virus (HIV), preserving cognitive performance in persons living with HIV (PLWH) has become a public health priority [5,6,7,8]. Given the synergistic impacts of neuroinflammation and vascular risk factors on the pathogenesis of premature cognitive decline in HIV, as well as the established role of olive oil in the prevention of the cognitive decline in persons living without HIV, it is plausible that olive oil may be beneficial to cognitive trajectories in PLWH [1,9,10]. We hypothesized that consumption of olive oil, shown in the literature to improve cognitive outcomes in persons living without HIV, would be associated with higher performance on cognitive tests among women living with HIV (WLWH). In this exploratory study, we sought to evaluate the association of olive oil with domain-specific cognitive performance (dCog) among a cohort of Chicago women living with and without HIV.

## 2. Materials and Methods

### 2.1. Participants

Participants were from the Cook County site of the Chicago Women’s Interagency HIV Study (WIHS), a longitudinal multicenter observational cohort study of HIV in US women. All participants provided written informed consent following approval of the site’s institutional review board. WIHS study methods and cohort characteristics are previously described [11,12]. From December 2015 to August 2016, Chicago WIHS Cook County site participants who consented to participate in a nested study of dietary intake completed the online version of the detailed Block 2014 Food Frequency Questionnaire (FFQ) with assistance from research staff [13]. WIHS participants complete a comprehensive cognitive battery every two years since 2009 [14]. This cross-sectional analysis included those women who had both Block FFQ and cognitive data collected within 18 months of FFQ. Of the 269 active Cook County WIHS participants, 166 had both the Block FFQ and cognitive testing assessed within 18 months and were therefore selected for analysis. The 166 women included (113 WLWH and 53 women living without HIV) were characteristically similar to the 103 Cook County WIHS women who were not included (data not shown). The race was stratified as White, African American, Hispanic, and Native American/Alaskan/Other.

### 2.2. Cognitive Measures

The WIHS comprehensive cognitive battery included the Hopkins Verbal Learning Test-Revised (HVLT-R), Stroop Test, Trail Making Test Parts A and B, Symbol Digit Modalities Test (SDMT), Letter Number Sequencing Test (LNS), Letter Fluency (F, A, S), where participants were asked to name as many words as possible beginning with the letters F, A, and S, as well as Semantic Fluency (animals) and Grooved Pegboard. A detailed description of the cognitive battery and the approach to scoring is described in previous WIHS publications [12].

The approach to computing dCog *t*-scores is detailed in prior publications [15]. Briefly, dCog *t*-scores were computed using a model that adjusts for potential practice effects in WIHS longitudinal retesting, age, education, race, and Wide Range Achievement Test (WRAT-3 Reading Recognition sub-score) [12,16]. The color-word conditions of the Stroop Test, Trail Making Test (TMT) Part B, and the working memory condition of the LNS were used to calculate the executive function dCog. The color-word condition of the Stroop test measures the ability to inhibit cognitive interference, where participants were required to read names of colors written in ink of a different color. TMT Part B measures cognitive flexibility, where participants were required to complete a sequence of ascending numbers and letters. The score for both Stroop and TMT was the time to completion of the task. For the LNS, participants were given a short-list of numbers and letters and instructed to repeat the numbers and letters in ascending order. The score was the total number correct.

The SDMT measures the time to pair abstract symbols with specific numbers. This test was used to determine psychomotor speed dCog. The score is the number of correctly coded items in 90 seconds. Attention/concentration performance was based on computed scores for Stroop Trials 1 and 2, TMT Part A, and the LNS Control/Attention Condition. TMT Part A requires to measure the time it takes for a participant to draw a line between numbers in ascending order. The average time to complete the Grooved Pegboard Test using the dominant and non-dominant hands was used to calculate the fine motor skills dCog. A letter and category fluency task, where participants were asked to name as many words as possible beginning with the letters F, A, and S and the total words generated in response to a category of animals, was used to calculate verbal fluency dCog. The score was the total words generated. HVLT-R was used to calculate verbal learning dCog. Single-trial learning and the total words recalled across each of the three learning trials were used to calculate the verbal learning dCog [14,15].

### 2.3. Dietary Evaluation

Included participants completed dietary evaluation within 18 months of cognitive testing. Participants were asked to report their usual frequency and quantity of intake of a variety of different foods and beverages (127 total items) over the past year using the Block 2014 FFQ [13]. The women were asked which fats or oils they primarily used for cooking or frying in their home. Women could select non-stick spray or none, butter, butter/margarine blend, stick margarine, soft tub margarine, low-fat margarine, olive oil, canola or saffron oil, vegetable oil, peanut oil, lard, vegetable shortening, and other oil. Women were divided into those who reported using olive oil and those who did not.

### 2.4. Covariates

Covariates, including potential confounders, were selected based on published WIHS neurocognitive literature [12] and included tobacco use (current, former, never), alcohol use (stratified as heavy- >12 drinks per week, moderate- 7 to 12 drinks/week, mild- ≤7 drinks/week, none), hard drug use defined as use of crack, cocaine and/or heroin (current, former, or never), marijuana use (current, former, and never), Hepatitis C (HCV) viremia defined as HCV RNA positivity on initial testing, and clinically significant depressive and post-traumatic stress disorder (PTSD) symptoms defined as a Center for Epidemiologic Studies Depression Scale (CES-D) score ≥ 16 [17] and PTSD Checklist Civilian (PCL-C) cutoff score of ≥44, respectively [18,19]. Current use, for both marijuana and hard drug covariates, was defined as any reported recent use of, former use was defined as use at baseline and no recent reported use, and never was defined as no report of substance use at baseline or recent visit. Additional clinical covariates included history of hypertension, defined as systolic blood pressure ≥ 140, diastolic blood pressure ≥ 90 or self-report or use of anti-hypertensive medications, hyperlipidemia, defined as low-density lipoprotein (LDL) ≥ 130 and/or their high-density lipoprotein (HDL) ≤ 40 mg/dL, diabetes, defined as ever self-reported anti-diabetic medication or any fasting glucose ≥126 or hemoglobin A1C ≥ 6.5% or self-reported diabetes, current body mass index (BMI = weight in kg/height in m^2^), where participants were categorized as underweight (BMI < 18.5), normal (18.5 to 25), overweight (BMI 25–30), or obese (BMI ≥ 30), and amount of moderate physical activity (past year frequency in minutes per day) calculated from Block 2014 FFQ responses. HIV-related covariates included nadir CD4+ T-cell count during WIHS observation or at time of highly active antiretroviral therapy (HAART) initiation and HIV plasma viral load. Covariates measured at the WIHS visit closest to FFQ administration were used in analyses.

### 2.5. Statistical Analysis

Linear regression models were used. The basic model adjusted for practice effect, age, education, race, and reading level (WRAT score). Associations in the basic model found to be significant (*p* ≤ 0.05) and underwent additional linear regression modeling to further evaluate the effects of covariates on these findings. Additional models included (1) basic model plus tobacco use, risky alcohol use, marijuana use, hard drug use, Hepatitis C viremia, and PTSD, (2) basic model plus cardiovascular risk factors, including hypertension, hyperlipidemia, diabetes, and moderate physical activity, and (3) basic model plus viral load and CD4 nadir (in HIV positive participants only). Analyses were performed using SAS (version 9.4, SAS Institute Inc, Cary, NC, USA).

## 3. Results

### 3.1. Demographic and Clinical Characteristics

Table 1 shows the demographic and clinical characteristics of the 113 WLWH and 53 women living without HIV included in analyses. The majority of participants in both HIV groups were African American (85%) with a median age of 50.7 (range 32.3 to 74.3 years of age) with no age or racial difference by HIV status. WLWH were less likely to have 13 or more (post-high-school) years of education (40.7% vs. 56.6%, *p* = 0.06); however, the mean WRAT-3 score did not differ by HIV serostatus and was 87.3 (sd 17.6).

The sample of women had high cardiovascular risk: 51.8% of the women were current smokers, 41.5% were obese, 54.2% had hypertension, 24.7% diabetes, and 22.6% hyperlipidemia with no significant differences in HIV status. Current hard drug use was reported in 11.5% of participants, and 56% reported previous hard drug use with no difference in HIV status. PTSD was reported in 18.9% of participants overall, but the occurrence was higher in WLWH. Median estimated moderate physical activity per day was 38 min for the study sample overall with significantly less activity in WLWH vs. women living without HIV (23.4 min vs. 127.8, *p* = 0.01).

### 3.2. Cognitive Performance of Women Living with and without HIV

dCog *t*-scores were similar among women living with and without HIV except for the domains of attention/concentration and verbal fluency; WLWH patients had significantly lower scores for attention/concentration (45.7 versus 52.0, *p* ≤ 0.0001) and verbal fluency (48.9 versus 52.9, *p* = 0.006).

### 3.3. Dietary Characteristics of Women Living with and without HIV

There were no significant differences in dietary intake between women living with and without, except for whole-grain intake. Diet quality overall among the sample of Chicago WIHS women was poor, with low intake of whole grains, berries, fish, green leafy vegetables and high intake of fast/fried foods, red meats, and sweets. For example, 71% of the study population reported eating less than one serving of whole grains per day. In the consumption of unhealthy food groups, greater than 18% reported consuming >2 servings per day of butter; more than a third reported consuming ≥7 servings of red meat per week; nearly 25% reported consuming ≥4 servings of fast/fried foods per week, and 66% reported consuming ≥7 servings of pastries and sweets per week. Nearly 25% of the participants reported using olive oil as their primary cooking oil. There was no significant difference in olive consumption between HIV groups (Table 1).

### 3.4. Association between Olive Oil and Cognitive Domain Performance

In the overall sample, women who reported using olive oil as their primary cooking oil scored 4.2 points higher in the attention/concentration domain performance compared to women who used another primary cooking oil (*p* = 0.02). In analyses stratified by HIV status, this association was only statistically significant in WLWH (estimate = 5.51, *p* = 0.01). Those who use olive oil also had higher verbal learning *t*-scores (estimate = 4.00, *p* = 0.02) and verbal memory *t*-scores (estimate = 3.91, *p* = 0.05). Both of these cognitive domain *t*-scores were only statistically significant for women living without participants (verbal learning: estimate = 6.49, *p* = 0.01; verbal memory: estimate = 8.12, *p* = 0.01) (Table 2).

The association of higher attention/concentration scores in WLWH using olive oil as their primary cooking oil maintained significance in all four adjusted models as follows: (1) basic model adjusted for practice effect, age, education, race, and reading level (WRAT score), (2) tobacco use, risky alcohol use, marijuana use, hard drug use, Hepatitis C viremia, and PTSD, (3) cardiovascular risk factors, including hypertension, hyperlipidemia, diabetes, and moderate physical activity, and (4) viral load and CD4 nadir (in WLWH only). The association for verbal learning and memory in women living without HIV was no longer significant after adjusting for tobacco use, risky alcohol use, marijuana use, hard drug use, Hepatitis C viremia, and PTSD (data not shown).

## 4. Discussion

Our cross-sectional study of the association between use of olive oil as a primary cooking oil and domains of cognitive performance in WLWH and sociodemographically similar women living without HIV suggests that olive oil use may be associated with higher attention and concentration in WLWH. The potential for dietary intervention in this patient population is of critical importance, particularly in this rapidly aging population. To our knowledge, this is the first study of the association between olive oil and cognition in WLWH.

Our analyses revealed differential associations between olive oil use and cognitive domains, specifically attention/concentration, verbal learning, and verbal fluency, in women living with and without HIV. WLWH, but not women living without HIV, who used olive oil as a primary cooking oil had higher cognitive *t*-scores in attention/concentration. Women living without HIV, but not WLWH, who used olive oil had higher verbal learning and verbal fluency scores. These differences between HIV subgroups suggest that diet mediates the effects of HIV on the brain.

Our findings in women living without HIV have been previously described. Previous studies of olive oil on cognition on persons living without HIV have found protective effects on verbal memory [1]. Our findings in women living without HIV for both verbal learning and verbal memory lost significance in a model adjusting for tobacco use, risky alcohol use, marijuana use, hard drug use, Hepatitis C viremia, and PTSD. It is possible that our findings were mediated by these factors rather than the olive oil itself.

In WLWH, we found using olive oil as a primary cooking oil was associated with higher attention/memory scores. HIV has been well described to affect attention/concentration [12,20,21,22,23]. Maki et al. found small but significant differences in attention between women living with and without HIV in the WIHS cohort [12]. Attention is of particular importance in this patient group as a decline in attention can have significant implications in the daily life of WLWH, affecting medication adherence and ability to complete activities of daily living [24]. Our study finding of higher attention/concentration scores in WLWH who use olive oil as their primary cooking oil may be indicative of potential neuroprotective effects of olive oil in this group. As there are no current treatments or preventative treatments for cognitive impairment in PLWH, the use of olive oil requires further study. However, we remain uncertain of why this group saw associations in attention/concentration but none of the other cognitive domains. Further study is needed to understand these differential associations.

Olive oil has a high content of monounsaturated fats, vitamins, minerals, and phenolic compounds, which are described to have antimicrobial, anti-inflammatory, and antioxidant effects [4]. In vitro and animal models, with olive oil, have demonstrated reductions in inflammatory markers, including interleukin-1beta (IL-1beta), interleukin-6 (IL-6), C-reactive protein (CRP), and matrix metallopeptidase 9 (MMP-9) [25]. The Mediterranean diet, with the addition of olive oil, has been shown to decrease CRP levels compared to a low-fat diet [26]. Oleic acid, found in olive oil, has been found to decrease the production of tumor necrosis factor-alpha (TNF-alpha) [27]. Inflammation has been strongly implicated in the pathogenesis of cognitive decline in PLWH [28,29,30]. The antioxidant and anti-inflammatory effects of olive oil may be leading to the differential protective associations between women living with and without HIV seen in our study. We had initially postulated that olive oil’s effects on cardiovascular disease could explain its neuroprotective associations with cognition. Though models adjusting for cardiovascular disease attenuated the change in cognition, in both women living with and without HIV, these findings maintained significance. This may support that there are multiple pathways, beyond the known cardiovascular benefit, by which olive oil can impact cognition in WLWH.

There are several limitations to this exploratory study. Firstly, this was a cross-sectional study, allowing only the assessment of associations but not causality. It is possible, for example, that participants with lower cognitive scores were less likely to use olive oil. The dietary evaluation was completed within 18 months of cognitive testing but could have been completed either before or after cognitive testing, again limiting the inferences of this study. We also did not have longitudinal dietary data, thus were unable to assess the impact of dietary changes on cognition over time or the long-term cognitive effects of olive oil use. Because our dietary data were obtained using a food frequency questionnaire, our data are susceptible to recall bias. Though we suspect a neuroprotective effect of phenolic compounds as a potential mechanism, our study did not differentiate the specific type of olive oil used. The sample size was small and consisted only of women, thereby limiting the generalizability of our findings. There were also several strengths in this study. As part of the WIHS cohort, research participants underwent extensive standardized cognitive testing across multiple cognitive domains. Additionally, our subjects completed the Block 2014 food frequency questionnaire, which provided extensive dietary information.

Our study suggests that attention/concentration was impaired in WLWH compared to demographically similar women living without HIV. Increased attention/concentration was associated with using olive oil as the primary cooking oil. This study suggests that the potential anti-inflammatory effects of olive oil may be beneficial for the cognitive trajectory of WLWH and highlights the urgent need for larger, longitudinal studies to articulate the extent of the impact of dietary components on cognitive function in PLWH.

## Figures and Tables

**Table 1 nutrients-11-01759-t001:** Characteristics of Block 2014 food frequency ^a^ Chicago WIHS (Women’s Interagency HIV Study) participants by HIV (human immunodeficiency virus) status. (*n* = 166).

Characteristics	Overall (*n* = 166) n (%)	WLWH (*n* = 113) n (%)	Women Living without HIV (*n* = 53) n (%)	*p* Value
Race/Ethnicity				
African American	141 (85)	95 (84.1)	46 (86.8)	0.14
Hispanic	12 (7.2)	6 (5.3)	6 (11.3)	
Other	2 (1.2)	2 (1.8)	0 (0)	
White	11 (6.6)	10 (8.8)	1 (1.9)	
Age, Median (IQR)	50.8 (13.3)	50.8 (12.4)	50.2 (13.3)	0.63
Years of Education, <13	90 (54.2)	67 (59.3)	23 (43.4)	0.06
Wide Range Achievement Test Score, Mean (SD)	87.3 (17.6)	86.0 (18.1)	90.1 (16.2)	0.16
Current Smoker	86 (51.8)	58 (51.3)	28 (52.8)	0.86
Risky Alcohol Use, >7 drinks/week	21 (12.7)	12 (10.6)	9 (17)	0.25
Current Hard Drug Use	19 (11.5)	11 (9.7)	8 (15.1)	0.31
Current Marijuana Use	36 (21.7)	24 (21.2)	12 (22.6)	0.84
Hypertension, Yes	90 (54.2)	62 (54.9)	28 (52.8)	0.81
Diabetes, Yes	41 (24.7)	31 (27.4)	10 (18.9)	0.23
Hepatitis C Virus, AB+/RNA+	19 (11.6)	14 (12.5)	5 (9.6)	0.59
Hyperlipidemia, Yes	33 (22.6)	23 (23)	10 (21.7)	0.87
High PTSD Symptom Burden (PCL-C ≥44)	31 (18.9)	28 (25.2)	3 (5.7)	0.003
Estimated Moderate Activity (min/day)				
Median (IQR)	37.8 (171)	23.4 (95.3)	127.8 (169.1)	0.01
HIV Viremia, Detectable		28 (24.8)		
Cognitive Domain *T*-Scores ^a^, Mean (SD)				
Executive Function	49.6 (11.3)	48.9 (12.3)	51.2 (8.9)	0.18
Psychomotor Speed	48.6 (9.3)	47.9 (9.7)	50.3 (8)	0.12
Attention and Concentration	47.8 (9.8)	45.7 (9.4)	52.0 (9.1)	<0.0001
Verbal Learning	51.4 (9.3)	50.7 (9.8)	52.8 (8.3)	0.20
Verbal Memory	50.6 (10.7)	49.9 (10.6)	51.9 (10.8)	0.26
Fine Motor Skills	47.9 (10.3)	47.3 (11)	49.2 (8.8)	0.28
Verbal Fluency	50.2 (9)	48.9 (9.1)	52.9 (8.2)	0.006
Olive Oil				
Primary oil used	39 (23.5)	25 (22.1)	14 (26.4)	0.54
Not primary oil	127 (76.5)	88 (77.9)	39 (73.6)	

^a^ Adjusted for baseline, second, third, later administration, age, education, race, WRAT (Wide Range Achievement Test). WLWH: Women Living with HICV; IQR: Interquartile Range; PTSD: Post-Traumatic Stress Disorder; PCL-C: PTSD Checklist Civilian.

**Table 2 nutrients-11-01759-t002:** Associations of olive oil and cognitive domain *t*-scores ^a^ among women living with and without HIV (human immunodeficiency virus).

Cognitive Domain	Overall, β (SE), Cohen’s d ^b^	WLWH β (SE), Cohen’s d ^b^	Women without HIV β (SE), Cohen’s d ^b^
Executive Function	3.48 (2.07), 0.31	3.97 (2.79), 0.33	2.25 (2.79), 0.26
Psychomotor Speed	2.51 (1.69), 0.27	2.85 (2.20), 0.30	1.57 (2.52), 0.20
Attention/Concentration	4.23 (1.78) *, 0.44	5.51 (2.12) **, 0.61	1.02 (2.87), 0.11
Verbal Learning	4.00 (1.69) *, 0.44	2.53 (2.23), 0.26	6.49 (2.44) **, 0.85
Verbal Memory	3.91 (1.94) *, 0.37	1.54 (2.42), 0.15	8.12 (3.20) **, 0.81
Fine Motor Skills	3.65 (1.91), 0.36	3.60 (2.54), 0.33	3.52 (2.74), 0.41
Verbal Fluency	0.89 (1.66), 0.10	1.43 (2.08), 0.16	–0.64 (2.58), –0.08

* *p* < 0.05, ** *p* < 0.01; ^a^ Adjusted for baseline, second, third, later administration, age, education, race, WRAT (Wide Range Achievement Test); ^b^ Cohen’s d effect sizes: small effect = 0.2; medium effect = 0.5; large effect = 0.8. WLWH: Women Living with HIV; SE: Standard Error.

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
