# Peer review of "Olive Oil Intake Associated with Increased Attention Scores in Women Living with HIV: Findings from the Chicago Women’s Interagency HIV Study"

_nutrients, 2019, doi:10.3390/nu11081759_

Round 1
Reviewer 1 Report
The paper by L.Warrior et al is a well written, interesting and worthy for publication.
One important point that could increase the usefulness of the paper is the information about the type of olive oil used as primary oil for cooking. The authors make an assumption about the role of the phenolic compounds (line 239) which is quite possible based on the available literature data, especially for oleocanthal or oleuropein aglycon and their neuroprotective activity (e.g https://pubs.acs.org/doi/10.1021/acschemneuro.9b00175, https://pubs.acs.org/doi/10.1021/acschemneuro.5b00190). However, those very interesting bioactive ingredients are found only in the virgin olive oil (especially in the extra virgin) and not in the refined olive oil. So, it would be really very useful if the authors know the type of used olive oil. If this information is available or can be retrieved, then it should be added. If not, then a relative comment about the potential role of those type of compounds should be added in the introduction and the discussion.
Author Response
Yes. We agree that knowing the type of olive oil would improve the study. Unfortunately that information was not collected. We have now added a comment about this to our limitations section. Thanks.
Reviewer 2 Report
Please see attached.

Round 2
Reviewer 2 Report
This manuscript is further improved. Title of Tale 2 needs to be changed to be consistent with the study design. This study is unable to estimate effects... only associations are possible.
This language of effects shows up in several places including discussion sections. The authors need to go through this consistently and make necessary modifications.
Author Response
This has been corrected in the text. Thank you.